# Nitrogen Addition Alleviates Microbial Nitrogen Limitations and Promotes Soil Respiration in a Subalpine Coniferous Forest

**Yang Liu** *,[†] **, Qianmei Chen** [†] **, Zexi Wang, Haifeng Zheng, Yamei Chen, Xian Chen, Lifeng Wang, Hongjie Li and Jian Zhang**

Long-term Research Station of Alpine Ecosystems, Key Laboratory of Ecological Forestry Engineering of Sichuan Province, Institute of Ecology & Forest, Sichuan Agricultural University, Chengdu 611130, China; 13880364175@163.com (Q.C.); wzxsolo@163.com (Z.W.); zhenghaifengsc@outlook.com (H.Z.); 15283521805@163.com (Y.C.);18227587142@163.com (X.C.); scwanglf@163.com (L.W.); lemontrain@163.com (H.L.); sicauzhangjian@163.com (J.Z.)

\* Correspondence: sicauliuyang@163.com or liuyang@sicau.edu.cn; Tel.: +86-199-4087-6810

† These authors contributed equally to this work.

**Abstract:** Soil microbes are an important component of soil ecosystems that influence material circulation and are involved in the energy flow of ecosystems. The increase in atmospheric nitrogen (N) deposition affects all types of terrestrial ecosystems, including subalpine forests. In general, alpine and high-latitude ecosystems are N limited. Increased N deposition could therefore affect microbial activity and soil respiration. In this study, four levels of N addition, including CK (no N added), N1 ($2\,\mathrm{g\,m^{-2}\,a^{-1}}$), N2 ($5\,\mathrm{g\,m^{-2}\,a^{-1}}$), and N3 ($10\,\mathrm{g\,m^{-2}\,a^{-1}}$), were carried out in a Sichuan redwood forest at the eastern edge of the Tibetan Plateau. The dynamics of soil respiration, major microbial groups, ecoenzymatic stoichiometry, and microbial biomass carbon and nitrogen (MBC and MBN, respectively) were investigated over a year. The results showed that N application significantly increased soil respiration (11%–15%), MBC (5%–9%), MBN (23%–34%), N-acetylglucosidase (56.40%–204.78%), and peroxidase (42.28%–54.87%) activities. The promotion of soil respiration, N-acetylglucosidase, and peroxidase was highest under the N2 treatment. The carbon, nitrogen, and phosphorus metabolism of soil microbes in subalpine forests significantly responded to N application. In the latter stages of N application, microbial metabolism changed from being N restricted to phosphorus restricted, especially under the N2 treatment. Soil bacteria (B) and gram-positive ($G^+$) bacteria were the dominant microbial groups affecting soil respiration. Structural equation modelling indicated that N application significantly promoted soil respiration and microbial biomass, whereas the main microbial groups did not significantly respond to N application. Therefore, we conclude that short-term N addition alleviates microbial nitrogen limitation and promotes soil respiration in the subalpine forest ecosystem that accelerates soil carbon (C) and N cycling. Continuous monitoring is needed to elucidate the underlying mechanisms under long-term N deposition, which may help in forecasting C, N, and P cycling in the alpine region under global climate change.

**Keywords:** ecoenzymatic stoichiometry; nitrogen addition; subalpine coniferous forest; soil microbial nitrogen restriction; soil respiration; soil microbial groups

## 1. Introduction

Atmospheric nitrogen (N) deposition is increasing in many ecosystems due to anthropogenic disturbances, such as fertilization and fossil fuel burning [1,2], which greatly affect soil biochemical properties [3]. China is becoming the world's third largest area of N deposition, which is attracting

increasing attention. Atmospheric N deposition on the eastern Tibet Plateau has tended to increase each year, and wet deposition has reached 7.55 to 12.84 kg N hm$^{-2}$ a$^{-1}$ [4] N deposition influences soil respiration, soil enzymes, and soil microbial biomass by regulating plant productivity and diversity and soil chemical and physical conditions [5–8]. For example, the most immediate effect of N deposition is NO$_3^-$ input, which can lead to soil acidification [9–11] and alter the soil organisms or trophic levels [12]. Although many studies regarding the effects of atmospheric N deposition on microbial respiration and microbial community composition in terrestrial ecosystems have been conducted [7,13–17], a clear understanding of microbial activity and microbial nutrient limitations in alpine regions remains elusive. A better understanding of soil microbial responses to N addition is critical to predicting the effects of global climate change on subalpine forest ecosystem processes.

Soil microbes play the main role in soil material cycling and energy flow. Atmospheric N deposition directly or indirectly affects the growth, reproduction, and activity of soil microbes and affects the structure and function of soil microbial communities by changing the soil's available N content, C/N, soil pH, litter decomposition process, etc. [18,19]. Soil respiration is associated with many components of ecosystem carbon cycles and plays a key role in regulating atmospheric carbon dioxide (CO$_2$) concentrations [20]. Furthermore, soils contain the largest active carbon pool on Earth. A small change in soil respiration may have a large impact on the net carbon fluxes [21,22]. Soil respiration is comprised of auto- and respiration [12,23], with heterotrophic respiration mainly originating from soil microbes. Soil respiration is thus an indicator of soil microbial activity, as these processes are closely connected. Soil microbial biomass is an important nutrient pool and source. In addition, atmospheric nitrogen deposition can directly or indirectly affect the main microbial groups and soil microbial activities, which will impact soil microbial biomass. Soil microbial biomass carbon and nitrogen (MBC and MBN, respectively) can thus be used as a measure of soil microbial activity under N deposition.

The theory of ecological stoichiometry focuses on how the elements of an ecosystem process are balanced when regulated by biological and environmental factors [24]. The stoichiometric ratio of extracellular enzyme activity characterizes the microbial demand for carbon, nitrogen, and phosphorus, and can be used an effective tool for detecting the environmental driving force of microbial metabolism and the availability of resources to ecosystems. N deposition affects soil microbial activities and soil microbial community structure by altering the accumulation of soil organic matter, which affects soil enzyme activity [25–27]. Furthermore, exogenous N can provide abundant nutrient sources for plants and microorganisms, thus affecting soil enzyme activity and changing ecosystem nutrient cycling [28–31].

At present, a global meta-analysis showed that atmospheric N deposition negatively affects soil microbial growth, composition, function, and respiration across all terrestrial ecosystems; these negative effects increased with the N application rate and experimental duration [32]. However, the responses of soil respiration and microbial activities to N addition were inconsistent in previous studies [26], showing promotion [13,33,34], inhibition [16,35], and no effects [5,36]. The response of major microbial groups to N in terrestrial ecosystems was found to be mainly influenced by the duration and content of N inputs [17,37]. In addition, different N deposition durations had inconsistent results, with long-term decreases [9,30,38] but short-term increases [33,34,39] among the main microbial groups. We hypothesize that differences in results and phenomena may mainly depend on ecosystem type. We suspect that atmospheric N deposition will alter the soil microbial activity and microbial nutrient limitation and further alter the material cycle in N-restricted subalpine forest. To assess whether and how N addition changes soil microbial activities and microbial nutrient limitation and further alters the material cycle in N-restricted subalpine forest, we conducted a field experiment to simulate N deposition in the subalpine coniferous forests of western Sichuan. Monthly soil respiration rates (RS), the identities of the main soil microbial groups according to phospholipid fatty acids (PLFAs), MBC, MBN, extracellular enzyme activities, soil soluble carbon (SC), soil dissolved inorganic nitrogen (DIN), and soil soluble phosphorus (SP) were measured. This study aims to improve our knowledge

regarding soil microbial metabolism and the carbon, nitrogen, and phosphorus balance in subalpine forest ecosystems.

## 2. Materials and Methods

### 2.1. Nitrogen Application Design

Sichuan redwood (*Larix mastersiana*) is an endemic species in China and is only distributed in some parts of Sichuan, especially in the alpine valley area of the Yangtze River basin. The experiment was conducted at the Alpine Forest Ecosystem Research Station, Miyaluo Nature Reserve, Sichuan Province, China ($31°43'$ N−$31°51'$ N, $102°40'$ E−$103°02'$ E). The experiment forest is located at 3600 to 3800 m a.s.l., with an area of approximately 6 hectares and forest canopy closure of 0.7. Five random plots (50 m × 50 m) were selected in the forest along the contour with the same aspect and similar slope, each treatment measuring 2.5 m × 2.5 m at about 5 m intervals (Figure 1). Plots were divided into four treatments (3 plots per treatment) as follows: (1) Blank control (CK: No N added), (2) low N (N1: 2 g m$^{-2}$ a$^{-1}$), (3) medium N (N2: 5 g m$^{-2}$ a$^{-1}$), and (4) high N (N3: 10 g m$^{-2}$ a$^{-1}$), which is in accordance with the actual N deposition that occurs in this region (7.55–12.84 kg N hm$^{-2}$ a$^{-1}$) (Zhu et al., 2015), representing N deposition increases of 2, 5, and 10 times in the N-treatment plots, respectively. The added N was initiated at the end of April 2017. Ammonium nitrate (NH$_4$NO$_3$) was applied monthly in 12 equal applications. In each application, the fertilizer was weighed, dissolved in 1 L of water, and applied to each plot using a solar automatic spraying device. The control plot received 1 L of water without fertilizer.

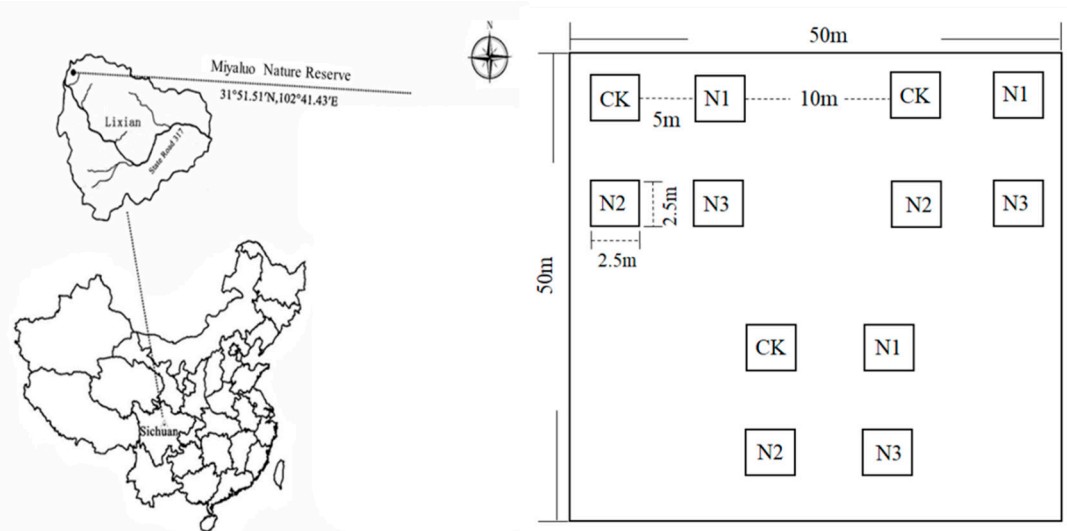

**Figure 1.** Map of the experimental plots and design. Control check, CK (blank control); N1 (low nitrogen treatment); N2 (medium nitrogen treatment); N3 (high nitrogen treatment).

### 2.2. Soil Respiration Measurement

The trench method was used to measure the soil heterotrophic respiration (Rh). In every N treatment, one cuboid of 1 m in length and width and 0.5 m in depth was dug and all roots were removed from the trench. Then, plastic film was placed around the trench to prevent root invasion, and the soil was then replaced. One polyvinylchloride (PVC) soil ring (inner diameter 20 cm, height 10 cm) was installed on the trench (5 cm in depth) to measure Rh, and another one was randomly installed outside the trench to measure the soil respiration rate (Rs). The CO$_2$ flux of the trenched quadrat PVC rings was treated as Rh, including soil faunal and microbial respiration, as well as CO$_2$ emitted by soil organic matter decomposition, while that in non-trenched plots was treated as the soil respiration rate (Rs). The difference of between Rs and Rh was assumed to be autotrophic respiration (Ra). On sunny days, measurement was conducted from 9:00 AM to 9:00 AM the next day for a total

of 72 soil rings (36 non-trenched for Rs + 36 trenched for Rh) using an Li–8100 infrared gas analyzer (Li–8100, Li–COR, USA) at the end of each month from June to October2017. A button temperature sensor (iButton DS1921-F5, Maxim/Dallas Semiconductor Inc., Dallas, TX, USA) was placed in a 10-cm soil depth in each plot to continuously monitor the temperature, and the sensor recorded data every two hours.

### 2.3. Soil Sampling and Analyses

Samples were collected at the end of each month from June to October 2017. In each plot, three soil samples from the 0–15 cm soil layer were randomly collected by using a 5-cm diameter soil drill. The samples were immediately refrigerated at 4 °C and shipped on ice to the laboratory, where they were screened for impurities and homogenized. These samples were sieved using a 2-mm sieve. Each sample was divided into three subsamples: The first was frozen at 4 °C for subsequent enzymatic activity and microbial biomass analysis, the second was frozen at −70 °C for phospholipid fatty acid analysis (PLFA), and the third was air dried for chemical analysis.

We identified the main microbial groups using phospholipid fatty acid (PLFA) analysis. The PLFA biomarkers were used to evaluate the fungal (18:2×6c, 18:1w9c, 20:4, 18:3) and bacterial populations (sum of 15:0, i15:0, a15:0, 16:0, 16:1×5t, 16:1×7c, 16:1×9c, 17:0, a17:0, i17:0, cy17:0, 18:1×7c, and cy19:0). The total microbial biomass was calculated according to the sum of all PLFA signatures detected. The $G^+$ bacterial population was calculated as the sum of i15:0, a15:0, i17:0, and a17:0, and the $G^-$ bacterial population was calculated as the sum of 16:1×7c, 16:1×9c, cy17:0, 18:1×7c, and cy19:0.

Enzyme activity was investigated using a partially modified version of the method developed by the Bob Sinsabaugh Lab [40]. The hydrolytic enzymes β-glucosidase (BG), β-N-acetylglucosaminidase (NAG), leucine aminopeptidase (LAP), and acid phosphatase (AP) as well as the oxidative enzymes polyphenol oxidase (PPO) and peroxidase (POD) were measured for absorbance using a microplate spectrophotometer. The stoichiometric ratios of certain enzymatic activities were also analyzed to gain a better understanding of possible resource shifts across treatments. The ratios lnBG:ln(NAG + LAP), lnBG:lnAP, and ln(NAG + LAP):lnAP provided information regarding enzymatic $C:N_{EEA}$, $C:P_{EEA}$, and $N:P_{EEA}$. Nutrient limitation was also measured using vector analysis (length, L; angle, A) of extracellular enzymatic activity (EEA) stoichiometry [41]:

$$\text{Vector L} = \sqrt{(\ln \text{BG}/ \ln [\text{NAG} + \text{LAP}])^2 + (\ln \text{BG}/ \ln \text{AP})^2}, \tag{1}$$

$$\text{Vector A} = tan((\ln \text{BG}/ \ln \text{AP}), (\ln \text{BG}/ \ln [\text{NAG} + \text{LAP}])) \times 180/3.1416, \tag{2}$$

where a relatively longer vector L indicates greater C limitation, and vector A < 45° and >45° indicate relative degrees of N- and P-limitation, respectively.

Soil microbial biomass carbon and nitrogen were monitored by chloroform fumigation extraction. Five grams of each soil sample were weighed and dried at 105 °C to measure the water content. Soil pH was measured at a soil-to-water ratio of 1:2.5. Distilled water was used to extract soluble C, N, and P, the samples were filtered by a 0.45-μm aqueous phase filtration membrane, and the soil soluble phosphorus (SP) was evaluated with the molybdenum antimony colorimetric method. Total organic carbon (TOC) (vario TOC cube/vario TOC select, Elementar Analysensysteme GmbH, Hanau, Germany) was used to monitor the soluble C and N according to the TIC/TNb method.

### 2.4. Data Processing and Statistical Analysis

The effects of N application treatment and sampling time on the soil respiration rate, microbial biomass, enzymatic activities, enzyme activity stoichiometric ratio, microbial relative nutrient limits, soluble carbon (SC), main microbial groups, dissolved inorganic nitrogen (DIN), SP, pH, and moisture content were analyzed by repeated measures ANOVA. The influence of N application on these indictors in every month were analyzed using least significant difference (LSD) multiple comparison.

Redundancy analysis (RDA) was performed using Canoco for Windows 4.5 to evaluate the relationships of soil respiration between MBC, MBN, the main microbial groups, and enzyme activities, and the main microbial groups between the environmental variables. All statistical analyses were performed in Excel 2010 and SPSS 22.0, and graphics were made with Origin 9.0.

Structural equation modelling (SEM) was conducted after classifying all variables into six groups, including soil chemical factors (pH, SC, DIN, SP, soil temperature (ST), and soil moisture content (SMC)), soil respiration (Rs, Rh, Ra), soil microbial biomass carbon and nitrogen (MBC and MBN, respectively), soil enzyme activity (AP, BG, CBH, NAG, LAP, PPO, and POD), and the main soil microbial groups (B, F, $G^+$, and $G^-$). Before the SEM analysis, principal components analysis (PCA) was performed to create a multivariate index representing each group to exclude autocorrelation among the variables. The first principal component (PC1), which explained 53.13% to 74.12% of the total variance in each group, was subsequently used in the SEM analysis. All included factors were subjected to logarithmic transformation to meet the assumptions of normality. The SEM analysis was conducted with the Amos 20.0 software package (Smallwaters Corporation, Chicago, IL, USA). The criteria for the evaluation of the structural equation model fit, such as *p*-values, $\chi^2$ values, goodness-of-fit index (GFI), and root mean square error of approximation (RMSEA) were adopted according to [42].

## 3. Results

### 3.1. Soil Respiration

As shown in Figure 2, N application significantly increased Rs and Rh ($p < 0.01$), Rs, Rh, and Ra significantly responded to sampling time ($p < 0.01$). The N2 treatment increased Rs more than N1 and N3, and the trend was similar for the Rh and Ra. Soil respiration was lowest in October.

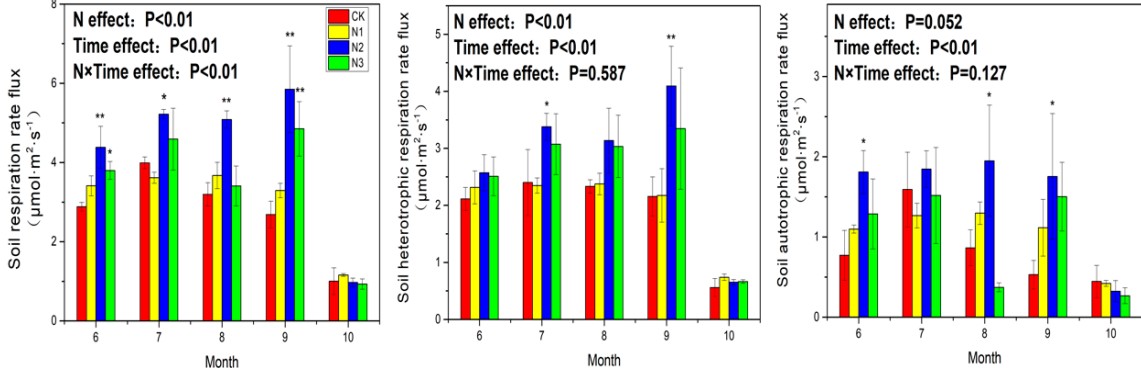

**Figure 2.** Responses of soil respiration to N application in a subalpine coniferous forest. * indicates a significant difference between N treatment and control ($p < 0.05$), ** indicates an extremely significant difference between N treatment and control ($p < 0.01$) (Repeated measures ANOVA, LSD multiple comparison).

### 3.2. Soil Microbial Biomass Carbon and Nitrogen

As shown in Figure 3, repeated-measures analysis of variance showed that N application had significant effects on the MBC and MBN ($p < 0.01$) but had no significant effect on the MBC:MBN ratio. The sampling time had significant effects on the MBC, MBN, and MBC:MBN ratio ($p < 0.01$). N application promoted the MBC and MBN in the growing season, but the trends differed between them. The MBC increased with increasing N concentration, with a significant increase in October. However, the MBN showed a trend of first increasing and then decreasing with increasing N level, and the MBN at every N application level was significantly higher than that in the CK. The MBC:MBN ratio was significantly lower than that in the CK in July and August.

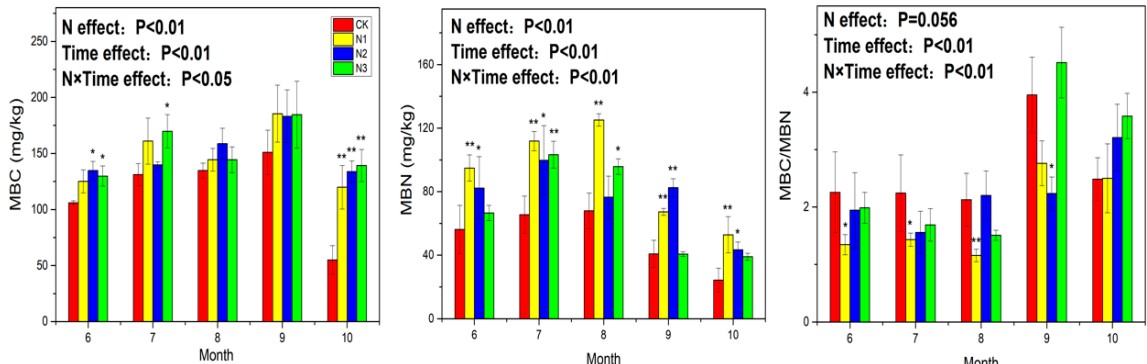

**Figure 3.** Responses of microbial biomass carbon (MBC), microbial biomass nitrogen (MBN), and MBC/MBN to N application in a subalpine coniferous forest. * indicates significant difference between N treatment and control ($p < 0.05$), ** indicates extremely significant difference between N treatment and control ($p < 0.01$) (repeated measures ANOVA, LSD multiple comparison).

### 3.3. Soil Enzyme Activity and Ecoenzymatic Stoichiometry

As shown in Figures 4–6, the activities of β-N-acetylglucosaminidase (NAG) and peroxidase (POD) in the soil, the soil ecoenzymatic stoichiometric ratio of $C:N_{EEA}$ and $N:P_{EEA}$, and the soil microbial relative nutrient limitation of vector L and vector A significantly responded to N application ($p < 0.05$). Soil enzyme activity, stoichiometric ratio and soil microbial relative nutrient limitation responded significantly to the seasonal dynamics ($p < 0.05$). The month with the highest soil moisture content was the month with the lowest activity of β-glucosidase (BG), polyphenol oxidase (PPO), and POD. In the latter stages of N application, soil enzyme activity increased at the N2-level treatment with more obvious promotion. During this process, the soil microbial metabolism shifted from N limitation to phosphorus limitation (Figure 6). Acid phosphatase (AP), BG, NAG, POD, and microbial relative N and phosphorus limits (vector A) significantly responded to the interaction between N application and sampling time ($p < 0.05$, $p < 0.01$).

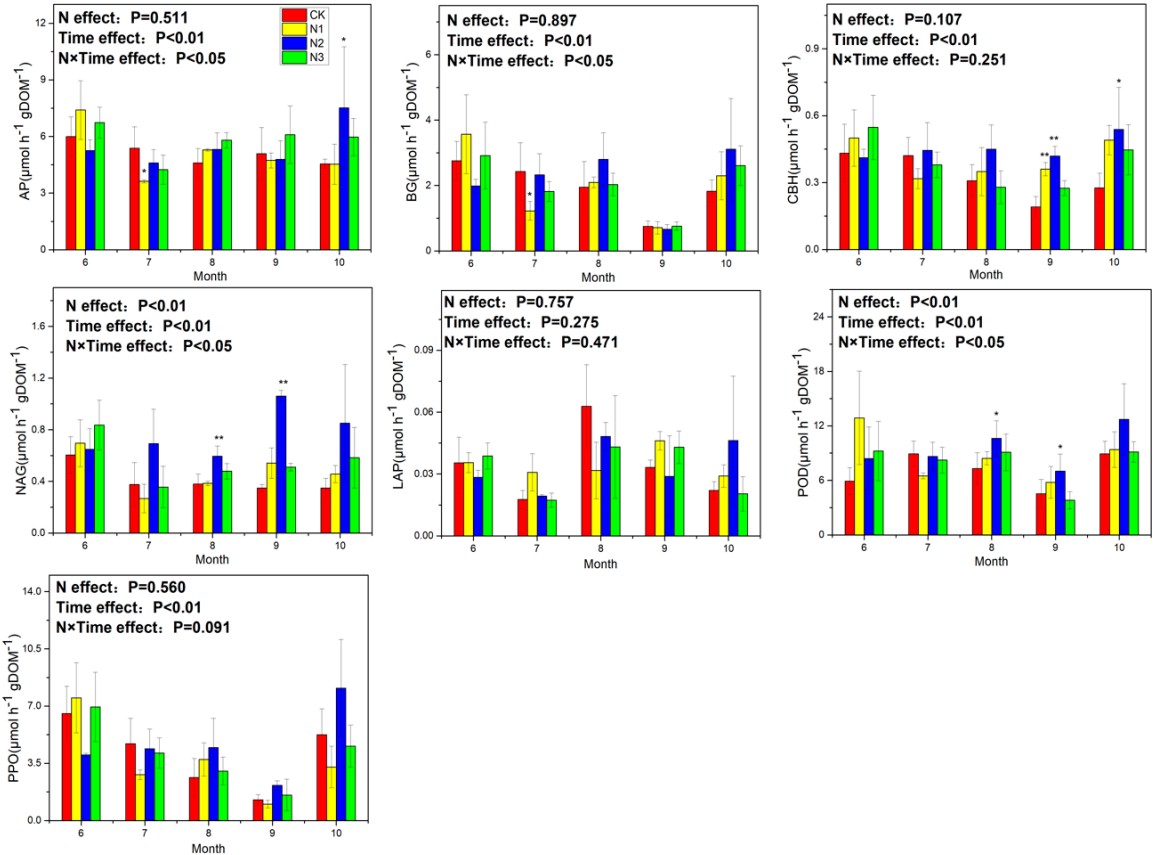

**Figure 4.** Response of soil enzyme activity to N application in a subalpine coniferous forest. AP, acid phosphatase; BG, β-glucosidase; CBH, cellobiohydrolase; NAG, β-N-acetylglucosaminidase; LAP, leucine aminopeptidase; PPO, polyphenol oxidase; POD, peroxidase. * indicates significant difference between N treatment and control ($p < 0.05$), ** indicates extremely significant difference between N treatment and control ($p < 0.01$) (repeated measures ANOVA, LSD multiple comparison).

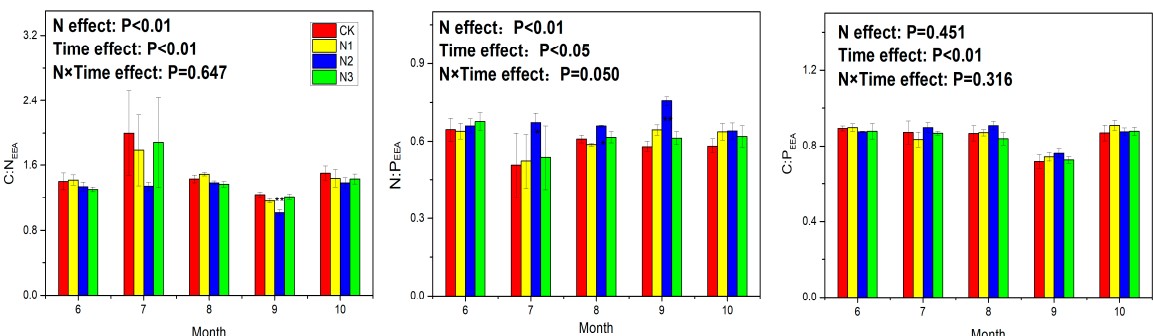

**Figure 5.** Response of soil enzyme activity stoichiometric ratio to N application in a subalpine coniferous forest. C:N$_{EEA}$, lnBG:ln(NAG + LAP); N:P$_{EEA}$, ln(NAG + LAP):lnAP; C:P$_{EEA}$, lnBG:lnAP.(repeated measures ANOVA, LSD multiple comparison).

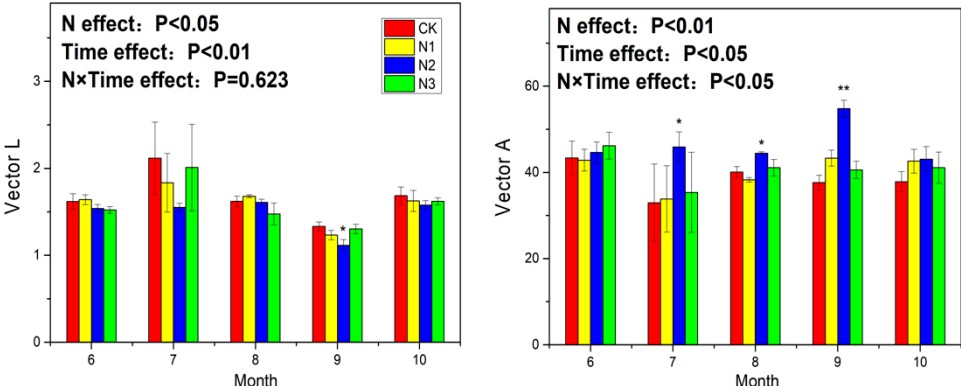

**Figure 6.** Responses of soil microbial relative nutrient limitation to N application in a subalpine coniferous forest. A relatively longer vector L indicates greater C limitation, and vector A < 45° and >45° indicate relative degrees of N and P limitation, respectively. * indicates significant difference between N treatment and control, ** indicates extremely significant difference between N treatment and control (repeated measures ANOVA, LSD multiple comparison).

### 3.4. Soil Physicochemical Properties

As shown in Figure S1, the daily average temperature of the 10-cm soil layer showed obvious seasonal fluctuations. The lowest daily average temperature was 0 °C, and the highest daily average temperature reached 12.7 °C. The temperature gradually increased from May to the end of August and then gradually decreased, reaching the lowest point in October. The soil water content was closely related to precipitation and showed obvious volatility.

Repeated measures ANOVA (Figure S2) showed that the SC, DIN, and SP did not significantly respond to N application, but SC and DIN significantly responded to sampling time ($p < 0.01$), and SC significantly responded to the interaction between N application and sampling time ($p < 0.05$). The SC and DIN gradually increased from June to July and then gradually decreased. There was no significant fluctuation in SP, and the N application did not significantly increase or decrease the SC, DIN, or SP. The soil pH value was relatively stable with its value of approximately 6.0. The soil pH value showed no significant response to simulated N deposition (Figure S2).

### 3.5. Soil Microbial PLFA Content

As shown in Figure S3, the soil microbial PLFA content did not significantly respond to N deposition but significantly responded to sampling time ($p < 0.01$), and bacteria significantly responded to the interaction between N application and sampling time ($p < 0.01$). The trends in bacteria and $G^+$ were the same, declining over time, while the trends in F, $G^-$, F/B, and $G^-/G^+$ first increased and then declined, with the highest values in August. In this study, the N2 treatment caused a significant increase in B and $G^+$ in June ($p < 0.01$), the N3 treatment significantly increased B and $G^-$ in October ($p < 0.05$), and the N addition treatment significantly decreased $G^-/G^+$ in September ($p < 0.05$).

### 3.6. Relationships among Variables

The RDA analysis showed that axes 1 and 2 explained 49.0% and 4.2% of the variation in the soil respiration rate, respectively (Figure 7a). $G^+$, B, MBN, and MBC were the dominant factors affecting soil respiration ($p < 0.01$) and explained 28.7%, 23.9%, 21.6%, and 20.4% of the variation, respectively. The RDA analysis further showed that axes 1 and 2 explained 26.8% and 7.3%, respectively, of the variation in the main microbial groups (Figure 7b). DIN, SC, SP, SMC, and ST were the dominant soil properties affecting the microbial community structure and explained 22.8%, 15.0%, 8.6%, 6.7%, and 5.0% of the variation, respectively. (Figure 7b; Table S1).

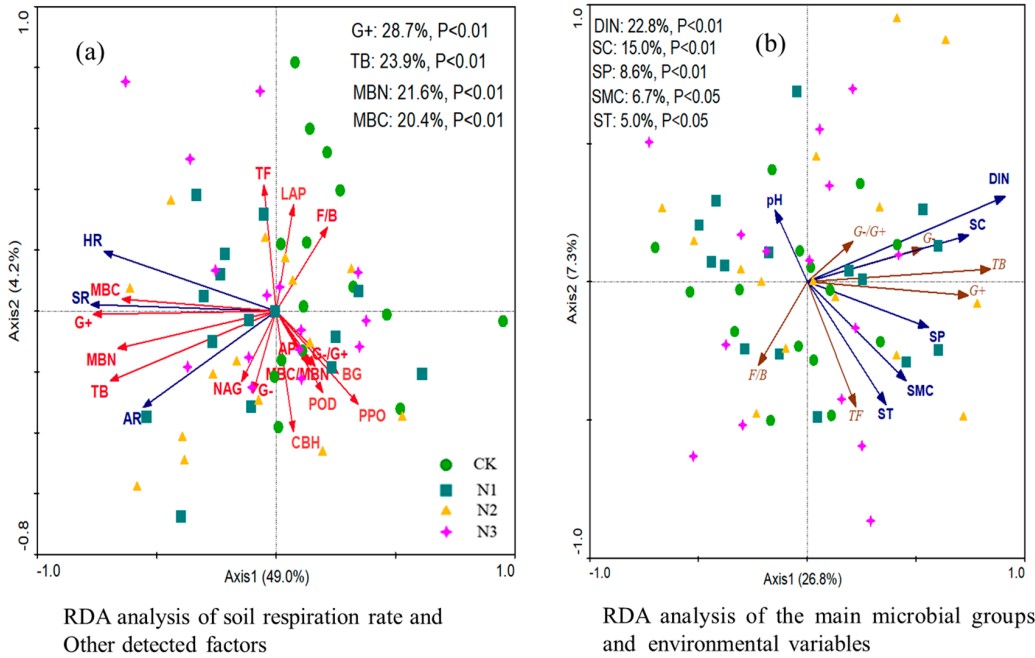

**Figure 7.** Redundancy analysis (RDA) of soil respiration between microbial N and microbial C. (**a**) Main microbial groups and enzyme activities; (**b**) main microbial groups and environmental variables.

The structural equation models fit the significance criteria according to their $x^2$, $p$, GFI, and RMSEA values ($x^2$ = 2.949, $p$ = 0.229, GFI = 0.995, and RMSEA = 0.051). The N application promoted soil respiration ($p < 0.01$), MBC, and MBN ($p < 0.05$); DIN, SC, and SP had a positive impact on the main soil microbial groups ($p < 0.01$), soil respiration ($p < 0.01$), and microbial biomass ($p < 0.05$); ST and SMC had a positive impact on the soil microbial biomass ($p < 0.01$); and soil microbial biomass had a positive impact on soil respiration ($p < 0.01$) (Figure 8, Table S2).

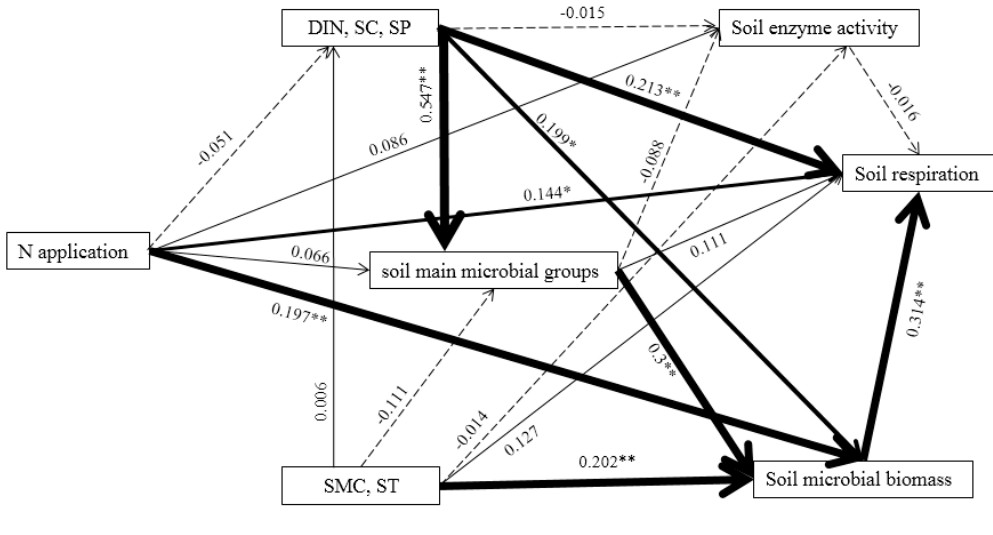

$$x^2 = 2.949, p = 0.229, \text{GFI} = 0.995, \text{RMSEA} = 0.051$$

**Figure 8.** Structure equation modeling depicting direct and indirect regulatory pathways of environmental factors that affect soil respiration, microbial biomass, enzyme activity, and soil main microbial groups at N application in the subalpine coniferous forest. Arrows represent positive (solid) or negative (dashed) path coefficients. Arrow width is proportional to the strength of the relationship. Numbers on the arrows are standardized direct path coefficients. * $p < 0.05$; ** $p < 0.01$.

## 4. Discussion

### 4.1. Response of Soil Respiration to N Application

We found that N addition strengthened the microbial activity in subalpine coniferous forest over a short period of time, and the soil respiration rate significantly increased by 11% to 15% under N application. Rs, Rh, and Ra increased to a greater extent under the N2 treatment than under the N1 and N3 treatments. There are three reasons for the N addition alleviating microbial N restriction. First, N deposition may increase root respiration [34]. Second, the soil nutrients of alpine ecosystems is N limited, and N addition increased N availability to stimulate the growth of soil microbial biomass and increased microbial respiration [14]. N addition effects on microbial biomass (G$^+$, B, MBC and MBN) were likely the most potent drivers of soil respiration, according to the RDA and SEM results. Third, increasing N availability benefited plant growth by enhancing new soil organic matter input, thus leading to an increase in soil respiration [16,43,44].

However, the effects of N application on soil respiration are inconsistent among different terrestrial ecosystems because of differences in vegetation type, soil microbial community, and other soil parameters [26]. For example, N addition decreased soil respiration by 11.2% to 17.8% from the subtropical to temperate forest but increased it by 7.9% in the cold temperate forest. The response magnitudes of soil respiration to nitrogen addition were all positively correlated with soil inorganic nitrogen content and soil pH value [45]. Water addition significantly enhanced ecosystem respiration while N addition had no effect on ecosystem respiration across three types of steppe [36]. A global meta-analysis showed that atmospheric N deposition negatively affects soil microbial growth, composition, and function across all terrestrial ecosystems, with more pronounced effects with an increasing N deposition rate and duration [32]. However, our results do not support this conclusion, and we found that the phenomenon mainly depended on the ecosystem type and duration of N application. A reason for the decrease in soil respiration is that the long-term N addition reduced the soil pH, limiting microbial activity and plant growth in the non-N restricted area [38,46]. Additionally, long-term N-rich soil conditions acclimate soil microbial communities to resist N inputs into the subtropical forest [47]. However, short-term N addition stimulated soil respiration, including heterotrophic respiration and autotrophic respiration, in two alpine coniferous forests [33]. In addition, a study showed that soil warming and N addition independently increased soil respiration and changed microbial community composition, as bacteria increased with N addition and protozoa increased with warming [13]. Others found that N addition over three years in an alpine meadow reduces soil respiration but increases the relative contribution of heterotrophic component [12]. On average, two years of N addition significantly reduced the annual soil respiration rate by 23.74% in a *Pinus tabuliformis* forest [16].

For the N addition levels, high levels of N addition significantly decreased soil respiration; however, low levels of N addition significantly increased soil respiration in subtropical forest [7], which was attributed to changes in the microbial biomass C and N, and soil substrate quality. High doses of N may affect the main microbial groups, resulting in inhibitory soil enzyme systems, which may be responsible for the decline in the soil respiration rate; this may also be due to some non-biological processes that immobilize a portion of the N in soil organic matter, which caused the slow rate of soil carbon release. The impacts of N deposition on soil respiration come to different or even opposite conclusions, in part because of the different ecosystem or forest types, and soil or climate conditions. We suggest that soil N availability may be vital to microbial physiology, and soil microbial communities may have different sensitivities in various ecosystems [48]. These factors should be considered for the simulation and prediction of the ecosystem carbon cycle in forests.

### 4.2. Response of Soil Microbial Biomass to N Application

Soil microbial biomass C:N:P stoichiometry are important parameters to determine the soil balance of nutrients and circulation of materials [31]. The results of this study showed that N application

significantly increased the MBC by 5% to 9% and the MBN by 23% to 34%, but that the MBC and MBN varied under different N application levels. The subalpine region of western Sichuan is an N-restricted area, and N application increases the available N in the soil over a short period of time, relieves the N limitation of microbes, and increases the fixation of N by microbes, thus significantly increasing the MBN. In comparison to CK, the N addition treatments decreased the MBC:MBN ratio, which suggests that there is a large potential for N release during microbial mineralization and conversion. We speculate that N application may change the requirement of soil microbes on C, N, and P, as well as affect soil organic matter (SOM) decomposition. However, the effect of N addition on soil microbial biomass has been found to be inconsistent; some studies found that N addition decreased soil microbial biomass [49,50]; recent research found that N addition first increased and then decreased the soil microbial PLFA content in a study conducted on the Loess Plateau [39]. Enhanced N deposition reduced soil microbial biomass and biomass respiratory efficiency in an oak forest [8]. N deposition has slowed fine root litter decay, and increased the contribution of lignin-derived compounds from fine roots to SOM [51]. When nutrient limitation was alleviated by fertilization, microbial biomass and enzymatic capacity for cellulose decomposition increased, which likely facilitates greater decomposition of soil organic matter [52]. Therefore, anthropogenic N deposition will change soil C storage in terrestrial ecosystems by affecting the microbial activity and composition.

However, in our study, N deposition did not significantly alter the main soil microbial groups (bacteria and fungi) and total amounts of PLFAs. Over the short term, the N inputs were not sufficient to induce soil acidification and the microbial accumulation of toxic metals to reduce the soil microbial biomass [37,53]. This may be the reason why the soil microbial community did not change significantly. However, a meta-analysis found N addition decreased both soil microbial diversity and microbial biomass [54], for a decreasing trend of bacterial biodiversity in Shannon and Simpson Chao1 indexes, especially the relative abundance of Actinobacteria and Nitrospirae, with greater declines caused by the higher dose of N addition. Long-term N deposition can negatively impact the soil microbial community [37]. However, some other results were inconsistent. Ramirez et al. [6] found that over the year-long soil incubation, N addition mainly increased the quantity of actinomycetes and depressed soil microbial activity by shifting the metabolic capabilities of soil bacterial communities in North America with relatively low contents of soil C and N [6]. Three years of urea fertilization significantly decreased soil bacterial diversity, whereas it increased fungal diversity and microbial biomass and respiration, through its negative effect on plant diversity in a semiarid grassland in China [17]. We consider that soil microbial communities respond in a different manner across ecosystems, and the N-mediated environment can play an important role in microbial diversity and functions, which appear to be mediated largely by ecosystem types and plant–soil interactions. Furthermore, the bacterial and fungal composition and microbial diversity under N addition should be further studied.

### 4.3. Response of Soil Enzyme Activity to N Application

The results of this study show that N application did not significantly affect soil enzyme activities except for those of NAG and POD, showing a promotion trend in the latter stages of N application, which was more obvious under the N2 level. One of the sources of soil enzymes is soil microbes, so short-term N application did not significantly alter the soil enzyme activities due to the occurrence of stable microbial PLFA biomass. Similar studies found that in stands of larch plantation in North China of varying ages, N addition did not significantly affect the activities of β-glucosidase (BG) and cellobiohydrolase (CB) [55]. Also, 4 to 5 years of nitrogen deposition had a minor effect on soil extracellular enzyme activities and ratios in six Chinese forests [56]. Soil enzyme activities significantly responded to sampling time due to obvious seasonal variation in air temperature and precipitation; thus, soil enzyme activity was more susceptible to external physical factors than to N application.

NAG and LAP are enzymes involved in N conversion and play an important role in the decomposition of nitrogenous substances in soil. Of these, NAG can degrade chitin and peptidoglycan, and LAP can hydrolyze proteins and amino acids. We found that the N2 treatment significantly

promoted NAG but not LAP activity in the latter stages of N application. Exogenous N addition may affect the decomposition of chitin and peptidoglycan, which accelerates the activity of NAG. However, N application may have less of an effect on soil protein and amino acids, and there was thus no significant effect on LAP activity. Some studies have found that N application can significantly suppress the activity of NAG and LAP [7,37,55]. In the case of less effective N in the soil, to break down organic N in the soil into available inorganic N, microorganisms produce NAG and LAP [55], and after exogenous inorganic N addition, the microorganisms no longer need to secrete excess NAG and LAP; thus, NAG and LAP were inhibited. The reason for the different results regarding soil enzyme activity may be related to forest type, tree species, litter composition, N application time, or salt toxicity.

PPO and POD are mainly involved in the decomposition of lignin, which is one of the main carbon sources on Earth. Cellulose and ligninase did not significantly respond to N addition, and these results were similar to those of [57]. However, in this study, BG, CBH, PPO, and POD all showed a promotion trend in the latter stages of N application, with that for the N2 treatment being the most obvious; thus, we suspect long-term N addition may promote cellulose and ligninase and accelerate the decomposition of cellulose and lignin to alter the carbon cycle on Earth. Ecoenzymatic stoichiometry showed that N application did not significantly affect carbon limitation, attributed to the minor impact on enzyme activity related to carbon conversion. However, other studies have shown that N addition can significantly promote [57,58] or inhibit [59,60] cellulose and ligninase. A meta-analysis of soil extracellular enzyme activities in response to global change found that N addition stimulated C-acquisition (9.1%) and P acquisition (9.9%) EEAs but suppressed oxidase activity (−6.8%), and EEAs are generally more sensitive to nutrient addition than to atmospheric and climate change [61]. Global environmental changes can alter EEAs, which has implications for soil carbon storage, nutrient cycling, and plant productivity. Further research is needed to elucidate the underlying mechanisms driving the responses of EEAs to global climate change.

### 4.4. Response of Ecoenzymatic Stoichiometry to N Application

Soil enzymatic stoichiometry (C:N$_{EEA}$, C:P$_{EEA}$, N:P$_{EEA}$) can provide a good indication of changes in microbial demands for energy and nutrients [24]. N application significantly affected C:N$_{EEA}$ and N:P$_{EEA}$ but not C:P$_{EEA}$, and seasonal changes also significantly affected the stoichiometric ratios of enzyme activity. In this study area, the C:P$_{EEA}$ value was 0.80 to 0.82, which was higher than the global average of 0.62, the C:N$_{EEA}$ value was 1.22 to 1.45, which was similar to the global average of 1.41; and the N:P$_{EEA}$ value was 0.58 to 0.67, which was higher than the global average of 0.44 [24]. These results indicated that the enzyme activity associated with C and N transformation in this region is high and that the enzyme activity involved in P transformation is low. According to the "best allocation" principle of ecological economics [24], microbes focus on a lack of resources, which reflects the lack of C and N more in alpine regions than in other regions. The lack of soil organic carbon was related to the low litter production in the Sichuan redwood forest and the thinness of the soil humus, which was consistent with the results regarding the soil soluble carbon (SC) content. Our findings highlight that the subalpine forest was originally N limited regarding the soil microbial relative nutrient limitation, and latter stages of N addition cause nutrient limitation transformation, from N limitation to phosphorus limitation. When the microbes obtain sufficient N at latter stages, phosphorus becomes the main factor limiting microbe metabolism. A similar study conducted in temperate grassland of northern China found similar results, as three-year N addition significantly increased AP and NAG activity, and shifted from nitrogen to phosphorus limitation [28], and N addition eased N-limitation but aggravated P-limitation of a meadow steppe in northeastern China [62]. In addition, organic N fertilization shifted the soil microbial biomass away from the excretion of N-degrading enzymes and toward the excretion of C-degrading enzymes. Further work on plant, soil, and microbial characteristics is needed to better understand the mechanisms of soil enzyme activities in response to N deposition in forest ecosystems.

## 5. Conclusions

Our study confirms the hypothesis that N addition promotes microbial activity (RS, MBC, MBN, NAG, and POD) and alters microbial nutrient limitations (C:$N_{EEA}$, N:$P_{EEA}$, vector L, and vector A) in a subalpine coniferous forest. These results indicate that the impacts of N deposition on soil microbial activity and the carbon, nitrogen, and phosphorus metabolism of soil microbes are ecosystem-type dependent. Soil microbial metabolism showed a particular pattern of N limitation in subalpine forests. N-deposition reaching level N2 (5 g m$^{-2}$ a$^{-1}$) could significantly increase the soil respiration and transformed the microbial metabolism from being N restricted to phosphorus restricted in the latter stages. Therefore, soil microbial activity will be promoted by short-term atmospheric N deposition in a subalpine forest ecosystem, but soil microbial activity and microbial nutrient limitation under long-term N deposition require further research and exploration. The mechanisms underlying the effect of simulated N deposition on soil respiration and microbial composition and nutrient limitation may help in forecasting C, N, and P cycling in alpine regions under future levels of reactive N deposition.

**Supplementary Materials:** The following are available online at http://www.mdpi.com/1999-4907/10/11/1038/s1, Figure S1: Dynamics of daily average temperature of 10 cm soil in the subalpine coniferous forests of western Sichuan. CK: blank control; N1: low nitrogen; N2: medium nitrogen; N3: high nitrogen. Figure S2: Response of soil soluble carbon, nitrogen and phosphorus, pH and moisture content to nitrogen application in the subalpine coniferous forest soils of western Sichuan. * indicates significant difference between nitrogen treatment and control, ** indicates extremely significant difference between nitrogen treatment and control (One-way ANOVA, LSD multiple comparison), Figure S3: Responses of soil microbial PLFAs content to nitrogen application in a subalpine coniferous forest. * indicates significant difference between nitrogen treatment and control ($p < 0.05$), ** indicates extremely significant difference between nitrogen treatment and control ($p < 0.01$) (One-way ANOVA, LSD multiple comparison). Table S1: Correlation coefficients between soil enzyme activity and its stoichiometric ratio, microbial PLFAs content, soil respiration rate, microbial biomass and soil physical and chemical factors ($n = 180$), Table S2: Correlation coefficients between soil respiration rate and soil enzyme activity, microbial PLFAs content, microbial biomass ($n = 180$).

**Author Contributions:** Y.L., Q.C., H.Z. and Y.C. designed the study; Q.C., Z.W., H.Z., H.L., and X.C. performed the experiments; Q.C., L.W. and Z.W. analyzed the data. All of the authors were involved in discussing the data. Y.L. and Q.C. drafted the manuscript, and all of the authors reviewed the manuscript.

**Funding:** This work was financially supported by projects from the National Natural Science Foundation of China (31570605), the Key Project of Sichuan Education Department (18ZA0393), the National Key Research and Development Plan (2017YFC0505003) and the Key Research and Development Project of Sichuan Province (18ZDYF0307).

**Acknowledgments:** We thank Shiyu Tang and Dedong Zhou for their great assistance in the fieldwork. We also acknowledge the reviewers for their constructive comments to improve the manuscript.

**Conflicts of Interest:** The authors declare no conflict of interest.

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
