# Peer review of "Nitrogen Addition Alleviates Microbial Nitrogen Limitations and Promotes Soil Respiration in a Subalpine Coniferous Forest"

_forests, doi:10.3390/f10111038_

Round 1
Reviewer 1 Report
Authors have addressed all comments. Thanks.
Reviewer 2 Report
Authors did a thorough revision. I do not see a need for further revisions (I found some minor typos, which however, can be fixed during proofreading). This is a really interesting study and I congratulate authors to their work.
This manuscript is a resubmission of an earlier submission. The following is a list of the peer review reports and author responses from that submission.
Round 1
Reviewer 1 Report
Liu et al investigated the impact of nitrogen addition on forest soil inhabiting microbial communities and tried to link it with microbial respiration. The manuscript offers interesting information and potentially contributes to the understanding of nitrogen effects on soil ecosystem properties; however, manuscript needs some improvements that may enhance the readership of this work.
Major comment:
Author compared their findings with a meta-analysis study, but they rarely tried to compare the effects of N addition in the context of other soil factors such as type of vegetation, or edaphic properties highlighted in that meta-analysis. Discussion section looks like introduction and results are not really discussed logically. Instead describing a comparison with past studies, authors need to describe some logical interpretation or discussion, reasoning behind their results. Authors heavily relied on old literature in describing N effects, but there are several new studies that help describe their results in a better way.
Minor:
Abstract:
Ln 12: change “ecological processes” to “soil ecosystems”
Ln 15-18. Instead of mentioning “meta-analysis” they say, previous research suggests that……
Ln 38. At the end of abstract, they may want to suggest to some future research directions or implications od their research.
Keywords: they may want to list some microbial functional groups here, e.g., gram-positive (G+) bacteria
Ln 53. One of the major consequences of N addition are soil acidification and alteration in soil organisms or trophic levels (e.g. " Functional ecology, 33(1), 175-187(2019; Ecology letters, 21(8), 1162-1173.).
Ln 54. Although many studies regarding……..any literature support or references????
Ln 81-84. Anthropogenic factors such as N addition may affect soil ecosystems, you may want to support your views with recent literature (Microbiome ecosystem ecology: unseen majority in an anthropogenic ecosystem. In Microbiome Community Ecology (pp. 1-11) along with (Kiss et al., 1998; Sinsabaugh, 83 2010).
Ln 91-95. You may want to support views with recent findings, along with Gallo et al., 2004.
Ln 138. “The nitrogen was added” instead of Nitrogen additions were initiated
The figure legends are stretched and does not look great.
Ln 252-253. “Soil enzyme activity and its stoichiometric ratio and soil microbial relative 252 nutrient limitation showed a significant response to the sampling time (P<0.05) with obvious 253 seasonal dynamics.” It is a confusing sentence, simplify it.
Ln 336. You may want to correct “(Zheng et al., 2018) (Guo et al., 2017).”
Reviewer 2 Report
Authors present an interesting study about N addition effects on soil CO2 efflux and microbial parameters in a high altitude forest in China. All parameters were measured between 1-6 months after starting the N application and therefore present rather short term treatment responses. However, the study is of high value because several reasonable N treatments were applied in a well replicated design. A larger set of interacting soil and microbial parameters were assessed. The structural equation modelling approach fits fine within the concept. Some improvements are required in the statistical analyses and the discussion needs to be refined. If authors successfully accomplish the suggested revisions, I believe that the paper could be a very valuable contribution to the journal.
General comments:
It is fine to test for treatment effects on soil respiration ect. by applying a repeated measures Anova. However, a post-hoc test needs to be used to distinguish between different treatment effects. So far only an overall N addition treatment effect (over all treatments N1, N2, N3) is presented (e.g. Fig2, 3 – here the significant differences between individual treatments could be indicated using different letters above the bars. This way a reader would know if there was a difference between treatment N1 and N2 and so forth). Such statistical results (F, p values…) could also be presented in a separate table if the figures become too crowded with the letters.
Figures need to stand alone. Therefore all the abbreviations on the axes need to be explained in the figure caption text (e.g. Enzyme fig, what does AP, BG… mean). Figure 6 is totally unclear as it is not explained what e.g. “Vector L” stands for. It is also unclear (this figure and text) what “soil microbial relative nutrient limitation” expresses. Please check all figures.
Abbreviations need to be used consistently. If for example nitrogen is abbreviated as “N” as was done in L15 already, then “N” needs to be used throughout the whole manuscript from L 15 onwards (and not a mixture of “N” and “nitrogen”!). The same, for sure, is true for all other abbreviations carbon, phosphorus... Please check and correct the whole text accordingly.
The discussion is not really convincing in parts, especially the section on soil respiration. Authors should try to work out what their findings actually mean. They applied 3 treatment levels and level N2 (and N3) showed the strongest responses whereas N1 (doubling of current N deposition) did not cause much response – what does this implicate for potential changes in N deposition in the region? How far must N-deposition increase that soil respiration becomes affected? (This would actually also be the main conclusion of the study – please adapt the conclusions accordingly). Also the rather short term nature of the results should be stressed a little more in the discussion section. What happened in other studies, did short term effects differ from long term effects there? There are many N application studies to compare this way. In the soil respiration chapter it should already be stressed that N addition effects on microbial biomass were likely the most potent drivers of soil respiration (as is nicely shown in structural equation modelling). Other chapters in the discussion (e.g. Biomass and plfas) are fine.
Specific comments:
Title is fine
Abstract is too long-winded:
You can delete all the text in L15-21. Just add a short sentence at L15 “…are nitrogen (N) limited. Increased N deposition therefore could affect microbial activity and soil respiration.” The next sentence would already be L21 “In this study…”. It is not necessary to present results of other studies (meataanalyses) and your hypotheses in the abstract.
L25: delete the unnecessary and repetitive parts of this sentence and just write “Nitrogen application significantly increased soil respiration…”
L27: add “activity” at the end of the sentence
L37: If you mention phosphorus here, then it has to be mentioned somewhere above (e.g. L24) that phosphorus, inorganic N and soluble C have been measured.
Intro is fine
L68: replace autotrophic by heterotrophic
Matreials
“2.1. Study area” can be totally deleted. It is not of any interest how the region around the study sites looks like. It is only of interest how the study sites look like (2.2.). Therefore please delete all the unnecessary information and integrate the relevant site climate and soil description into chapter 2.2.
All the other material sections are fine
Results and Discussion
As mentioned above, here a post hoc test would help to clarify which treatment had which effects, especially with regard to soil respiration (whay one way Anova in Fig 2? – in the text it is written that soil respiration was tested by repeated measures Anova…)
Total amounts of PLFAs were not different among treatments, but MBC was different. This might be discussed a little. In the discussion it should also be stressed that basically MBN was elevated in the N application treatments.